# Transferring Labels to Solve Annotation Mismatches Across Object Detection Datasets

**Yuan-Hong Liao**[2]*, **David Acuna**[1], **Rafid Mahmood**[1,3], **James Lucas**[1], **Viraj Prabhu**[4], **Sanja Fidler**[1,2]

[1]NVIDIA, [2]University of Toronto, Vector Institute, [3]University of Ottawa,
[4]Georgia Institute of Technology,
andrew@cs.toronto.edu

## Abstract

In object detection, varying annotation protocols across datasets can result in *annotation mismatches*, leading to inconsistent class labels and bounding regions. Addressing these mismatches typically involves manually identifying common trends and fixing the corresponding bounding boxes and class labels. To alleviate this laborious process, we introduce the label transfer problem in object detection. Here, the goal is to transfer bounding boxes from one or more source datasets to match the annotation style of a target dataset. We propose a data-centric approach, Label-Guided Pseudo-Labeling (LGPL), that improves downstream detectors in a manner agnostic to the detector learning algorithms and model architectures. Validating across four object detection scenarios, defined over seven different datasets and three different architectures, we show that transferring labels for a target task via LGPL *consistently* improves the downstream detection in every setting, on average by 1.88 mAP and 2.65 $AP^{75}$. Most importantly, we find that when training with multiple labeled datasets, carefully addressing annotation mismatches with LGPL alone can improve downstream object detection better than off-the-shelf supervised domain adaptation techniques that align instance features.[1]

## 1 Introduction

Supervised learning via large carefully annotated datasets are pivotal for object detection applications such as autonomous driving (Feng et al., 2019). A natural strategy for creating a large dataset is to mix multiple different datasets. However, different datasets have different annotation protocols which specify dataset-specific interpretations of accurate class labels and bounding boxes. These differing protocols result in *annotation mismatches* when mixing multiple datasets to train a model, which consequently hampers downstream performance.

Annotation mismatches are pervasive between object detection datasets. While prior works addressing image domain mismatches (Acuna et al., 2022; 2021; Li et al., 2022; Yao et al., 2021; Wu et al., 2022) are widely explored, annotation mismatches are relatively less discussed (Wang et al., 2020; Wood et al., 2021), especially in object detection. In Fig. 1 (left), Mapillary Vistas Dataset (MVD) (Neuhold et al., 2017) annotates cyclists as 'riders', while Waymo Open Dataset (Waymo) (Sun et al., 2020) combines riders and bicycles into the 'cyclist' class. On the other hand, nuImages (Caesar et al., 2020) annotates bikes on sidewalks, but Waymo excludes these per the annotation instructions. In addition to the ontological mismatches, discrepancies of annotation instructions, human-machine misalignment, and cross-modality labels result in unique annotation mismatches.

This work characterizes four types of annotation mismatches and introduces a generic algorithm that addresses the annotation mismatches and consistently enhances the downstream performance of several object detectors. Given a source dataset used to augment a target dataset, our goal is to train a label transfer model to modify the source labels so that the source labels are more *target-like*, *i.e.* following the target annotation protocols. Label transfer aligns the annotation mismatches in a data-centric manner and can serve as a pre-processing step in the existing training workflow, regardless of the downstream detector architectures and learning algorithms as shown in Fig. 1 (right).

---

*Work done while Yuan-Hong Liao was an intern at NVIDIA
[1]Project website will be at: https://andrewliao11.github.io/label-transfer

Figure 1: **Left:** Varying annotation protocols across datasets can result in *annotation mismatches*, leading to inconsistent labels. For example, MVD Neuhold et al. (2017), nuImages Caesar et al. (2020), and Waymo Sun et al. (2020) disagree with what a cyclist represents. Yellow dashed bounding boxes are not annotated. **Right:** Label transfer is a *data-centric* approach that transfers the labels from one dataset to match another dataset's annotation protocol, which can be considered as a pre-processing step in the existing training workflow.

We find that intuitive and existing data-centric strategies for our new problem generally do not leverage all available information. For example, we may use a model trained on the target data to generate pseudo-labels on the source images (Arazo et al., 2019; Lee, 2013), but this discards the existing source labels. On the other hand, statistical normalization (Wang et al., 2020) aligns boxes statistics but ignores the image content. However, both practices leverage only partial information from the source dataset, which we show can lead to suboptimal results.

We propose a data-centric approach, Label-Guided Pseudo-Labeling (LGPL), that leverages full information in the source dataset to generate consistent and accurate transfer. LGPL repurposes the standard two-stage object detector architectures for label transfer. We quantify the effectiveness of our label transfer model by training a detector on the combination of the target dataset and the label-transferred source dataset and evaluating on the target domain. Our contributions include: (1) We formalize the label transfer problem and propose a taxonomy characterizing the annotation mismatches across object detection datasets. (2) We develop a data-centric algorithm, LGPL, that extends the concept of pseudolabeling by leveraging source dataset bounding boxes and class information for label transfer. (3) We validate our approach on four transfer scenarios across seven datasets. LGPL consistently outperforms baselines in every scenario, whereas baseline approaches sometimes perform worse than simply not transferring labels at all. Finally, we show LGPL outperforms off-the-shelf supervised domain adaptation (Prabhu et al., 2023), showing that our data-centric framework is a successful alternative to model-centric strategies for addressing data distribution shifts.

## 2 RELATED WORK

**Image distributions misalignment.** Approaches such as MMD (Yan et al., 2017), domain adversarial learning (Hoffman et al., 2018), and self-training (Li et al., 2022) align the image distributions during training. Our work focuses on aligning annotation mismatches in a data-centric manner, which can be considered as a pre-processing step in the existing training workflow. Similarly, Arruda et al. (2019) adopts image translation as a pre-processing step to align image distributions.

**Label space misalignment.** Combining heterogeneous label spaces over multiple datasets requires careful manual splitting and merging (Lambert et al., 2020). Zhou et al. (2021) avoid the laborious manual work by learning a common label spaces for all datasets considered. Recently, Chen et al. (2023); Meng et al. (2023) use language as a bridge between diverse object detection datasets (Radford et al., 2021; Ilharco et al., 2021) by learning a label space that encourages positive transfer between datasets. Our work differ with the above multi-dataset detection approaches in: 1) We consider scenarios where the class label spaces are matched but the annotation protocols are different, therefore, leading to annotation mismatches (see Section 3). 2) We optimize the target performances, while multi-dataset detection optimizes the average performances of all datasets considered. This work shows that even when the class label spaces are well-aligned, leveraging datasets from different sources might still suffer from annotation mismatches.

**Annotation mismatches.** Factors such as how data is collected or specific instructions given to annotators can lead to idiosyncratic annotations for different datasets. Beyer et al. (2020); Yun et al. (2021); Recht et al. (2019) address these undesirable properties in ImageNet through careful

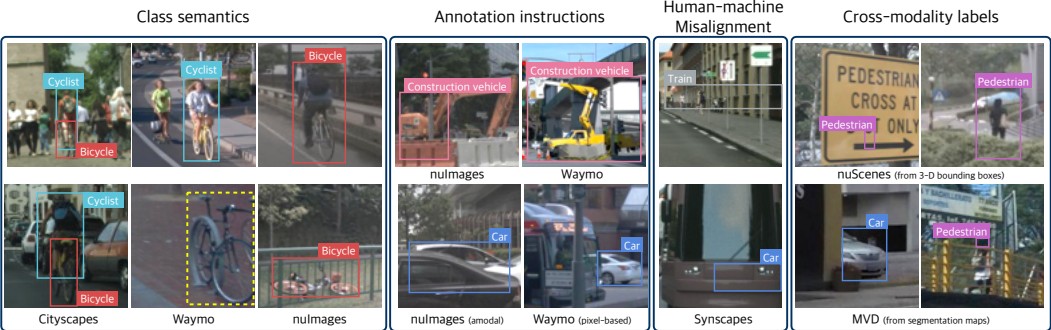

Figure 2: **Examples of annotation mismatches.** We identify four factors contributing to annotation mismatches. Multiple factors can influence annotation mismatches within a dataset. Yellow dash bounding boxes are not annotated.

scrutiny and re-annotations. Rottmann & Reese (2023); Wood et al. (2021) smooth the labels and perform label adaptation to close the gap between synthetic and real-world semantic segmentation datasets. Our work focuses on addressing the annotation mismatches in object detection datasets, which involves producing class labels and bounding regions for every instance. We provide a detailed explanation of annotation mismatches in the context of object detection datasets in Section 3.

## 3 TAXONOMY OF ANNOTATION MISMATCHES IN OBJECT DETECTION

In this section, we pinpoint four fundamental types of annotation mismatches observed in five real-world and two synthetic datasets.

**Class semantics.** Datasets can differ in the class ontology, meaning an object can be left unlabeled or labeled as a different class depending on the dataset. For example, Waymo (Sun et al., 2020) considers a 'bicycle' (or equivalently 'cyclist') as a combination of a bicycle and a rider, excluding parked bicycles. In contrast, nuImages and nuScenes (Caesar et al., 2020) annotate all visible bicycles. Additionally, differences in class semantics can lead to variations in labeling. For instance, Waymo combines the bicycle and rider into a single bounding box, while Cityscapes (Cordts et al., 2016) assigns separate classes for 'bicycle' and 'cyclist'.

**Annotation instructions.** Annotators may receive specific instructions that disagree between datasets. For example, in Waymo, construction vehicle bounding boxes include hydraulic arms, while nuImages omits them, leading to smaller bounding boxes. Moreover, nuImages annotators were instructed to estimate occluded parts, producing amodal bounding boxes (Nanay, 2018), whereas Waymo annotations are limited to visible pixels.

**Human-machine misalignment.** Automatically annotated synthetic image datasets, constructed using 3-D simulated scenes, exhibit discrepancies compared to human-annotated datasets. For instance, on Synscapes (Wrenninge & Unger, 2018), objects that are heavily occluded, truncated, or distant from the camera are perfectly annotated, which does not occur in human annotations. Consequently, learning from synthetic datasets with abundant occluded annotations can lead to object detector hallucinations.

**Cross-modality labels.** Real datasets may also utilize auxiliary modalities for annotation. For example, 3-D datasets can be converted to 2-D by projecting 3-D bounding boxes (*e.g.*, nuScenes), and segmentation annotations can be used to automatically produce bounding boxes (*e.g.*, Cityscapes, MVD (Neuhold et al., 2017)). However, integrating side information can introduce idiosyncrasies, including pixel-based bounding boxes that ignore occluded regions (see Annotation instructions above) or the production of oversized, occluded, or truncated boxes due to object geometry and changes in viewing perspectives during 2D projection of 3D scenes.

Combining multiple datasets may yield a mixture of annotation mismatches. We analyze the severity of annotation mismatches in Appendix A.3. Rather than developing custom solutions to address each mismatch individually, we focus on developing a solution for learning to address any common annotation mismatch trend between two object detection datasets.

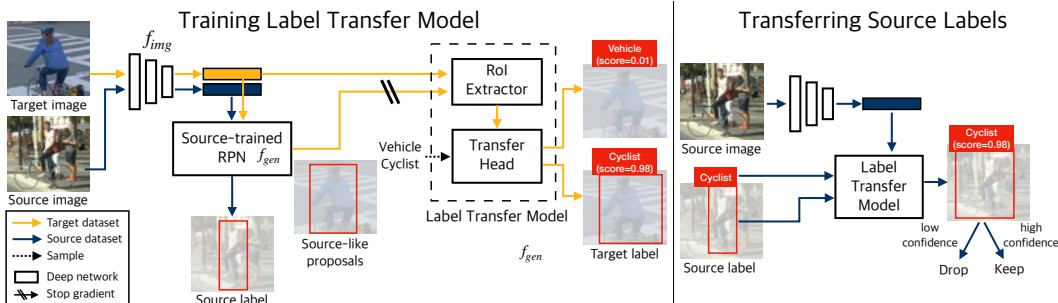

Figure 3: **Label-Guided Pseudo-Labeling** consists of an image encoder $f_{img}$, a box generator $f_{gen}$, and a label transfer model $f_{trans}$. **Left:** We train the box generator on the source dataset to generator source-like bounding boxes, and train the label transfer model on the target dataset to capture the target annotation biases. **Right:** During inference, the label transfer model takes the source labels as inputs. We construct the label-transferred source datasets via thresholding.

## 4 METHOD

We propose Label-Guided Pseudo-Labeling (LGPL), a data-centric algorithm. LGPL extends the concept of pseudo-labeling and addresses any annotation mismatch between two object detection datasets. We first introduce the notation and the label transfer problem (Sec. 4.1) and provide background knowledge (Sec. 4.2). Finally, we present LGPL in Section 4.3.

### 4.1 PROBLEM FORMULATION

**Notation.** Let $\mathcal{X}$ and $\mathcal{Y}$ denote an input and output space, respectively. In $K$-class object detection, the input are images $x \in \mathcal{X}$ and the output is a set of $N$ annotated objects $y := (b, c)$ comprised of 2-D bounding boxes $b \in \mathbb{R}^{N \times 4}$ and class labels $c \in [K]^N$, where $g : \mathcal{X} \to \mathcal{Y}$ denotes a potentially noisy ground truth labeling function that depends on the annotation protocols of a specific dataset (*e.g.* semantics, annotator instructions, auxiliary modalities).

**The label transfer problem.** Suppose we are given two datasets, a source $\mathcal{D}_{src} := \{(x, g_{src}(x))\}$ and target $\mathcal{D}_{tgt} := \{(x, g_{tgt}(x))\}$, of input-output pairs defined by two different ground truth labeling functions $g_{src}$ and $g_{tgt}$, respectively. Without loss of generality, we assume that the datasets have different images but share the same class label space $[K]$, i.e., if the class label spaces differ, then we only need to consider the common classes.

A label transfer model is a function $f_{trans} : \mathcal{X} \times \mathcal{Y} \to \mathcal{Y}$ that takes as input a image-label pair from the source dataset and generates a bounding box and a class label that matches the target labeling function, *i.e.*, in an ideal setting $\forall x \in \mathcal{D}_{src}, f_{trans}(x, g_{src}(x)) = g_{tgt}(x)$. After training, a label transfer model can map the source dataset to a new label-transferred dataset whose labels are 'target-like', which we then use to augment the target dataset and train a downstream object detector. However, the main challenge in training the label transfer model is the lack of paired supervision, *i.e.* a training dataset of triplets $(x, g_{src}(x), g_{tgt}(x))$. We propose LGPL in Sec 4.3 to address this.

In this work, we assume that (1) the class labels are the same between the source and the target labeling functions and (2) the source labeling function either detects or over-detects all the objects specified in the target labeling function. This reflects our observations from the common annotation mismatches, (*e.g.* transferring labels from a simulated dataset to a real-world dataset, see Section 3 and Section A.3 for our empirical validation). Under this assumption, label transfer simplifies to two sub-problems: given bounding boxes and class labels for a source dataset image, (1) determine if each object would be labeled under the target annotation protocol; and (2) determine an appropriate target-like shape of the bounding box.

### 4.2 PRELIMINARIES ON TWO-STAGE OBJECT DETECTION

We first briefly review two-stage object detection algorithms as our label transfer algorithm will later leverage this framework. A two-stage object detector (*e.g.* Faster-RCNN (Ren et al., 2015),

Cascade-RCNN (Cai & Vasconcelos, 2018)) consists of an image encoder $f_{\text{img}}$, a region proposal network (RPN) $f_{\text{RPN}}$, and a region-of-interest (RoI) head $f_{\text{RoI}}$. In the first stage, the RPN learns to generate bounding box candidates from extracted image features via an RPN loss $\mathcal{L}_{\text{RPN}}$ that combines of a regression loss between anchors and predicted boxes with a binary cross entropy loss for correctly proposing a region. In the second stage, the RoI head learns to refine these bounding box candidates and classify the object in the region via an RoI loss $\mathcal{L}_{\text{RoI}}$ that combines a $K$-way cross entropy loss and a regression loss between ground truth and candidate boxes. By chaining these two networks with the image encoder, we can input an image to predict a set of bounding boxes and class labels.

### 4.3 LABEL-GUIDED PSEUDO-LABELING

The main challenge of label transfer is the lack of paired supervision. We now introduce our label transfer algorithm, which relies on the observation that an RPN trained on a given dataset will generate RoIs resembling the annotation style of the given dataset. For example, if a dataset labels cyclists to include the rider and bicycle, then the RPN will generate regions covering both. Drawing on this observation, we re-purpose the RPN and RoI head as a box generator $f_{\text{gen}}$ and a label transfer model $f_{\text{trans}}$, respectively. For the sake of brevity, we extend the notations by allowing $f_{\text{gen}}$ and $f_{\text{trans}}$ to take images as input, leveraging the shared image encoder $f_{\text{img}}$. The box generator learns to generate source-like bounding regions for the target dataset. The label transfer model learns to (1) map source-like regions to the corresponding target bounding boxes and (2) determine the validity of the bounding regions. At inference time, we discard the box generator and use the label transfer model to directly map source labels to their target style. Fig. 3 summarizes this workflow.

**Learning to create source-like bounding boxes.** To learn to map from source-like to target-like boxes via supervised learning, we require a training dataset of triplets $(x, g_{\text{src}}(x), g_{\text{tgt}}(x))$. Since this is unavailable from either of our two datasets, the box generator synthesizes source-like proposals $f_{\text{gen}}(z)$ that approximate the bounding boxes of the $g_{\text{src}}(x)$ for the target dataset. The box generator can be trained simply by the standard RPN loss over the source dataset only.

**Learning to transfer labels.** Recall from our main assumption that label transfer does not require generating a correct class label, but only determining whether the object should be labeled as well as the corresponding bounding box. To train this transfer model, we use triplets $(x, y', g_{\text{tgt}}(x)|y' = [f_{\text{gen}}(x), c])$ where $c \sim [K]^{|f_{\text{gen}}(x)|}$ is a set of randomly sampled class labels, for all $x \in \mathcal{D}_{\text{tgt}}$. Assigning these random class labels to each proposal in training ensures that $f_{\text{trans}}$ learns to correctly determine whether the region should be labeled (*e.g.* rectifying annotation mismatches from highly-occluded objects).

The label transfer model is architecturally different from the conventional RoI head, as our new head must receive class labels concatenated with box features. Moreover, inspired by two-stage object detectors, our model sidesteps the combinatorial complexity of the set-to-set problem of predicting multiple labels for an image. Here, we employ a class-conditional assigner, which assigns a training target to each input detection label by matching the class label and comparing the IoU. This assigner is useful in crowded scenes where multiple objects can overlap with the candidate bounding box.

**Training and inference.** We summarize the training of $f_{\text{img}}$, $f_{\text{gen}}$, and $f_{\text{trans}}$ in LGPL as

$$f_{\text{img}}^*, f_{\text{gen}}^*, f_{\text{trans}}^* \leftarrow \underset{f_{\text{img}}, f_{\text{gen}}, f_{\text{trans}}}{\arg\min} \sum_{x,y \in \mathcal{D}_{\text{src}}} \mathcal{L}_{\text{RPN}}(x, y, f_{\text{img}}, f_{\text{gen}}) + \sum_{x,y',y \in \mathcal{D}_{\text{trans}}} \mathcal{L}_{\text{RoI}}(x, y', y, f_{\text{img}}, f_{\text{trans}})$$

$$\mathcal{D}_{\text{trans}} \leftarrow \{(x, [\text{StopGrads}(f_{\text{gen}}(x)), c], y)|c \sim [K]^{|f_{\text{gen}}(x)|}, (x, y) \in \mathcal{D}_{\text{tgt}}\} \tag{1}$$

where $\mathcal{L}_{\text{RPN}}$ is the standard RPN loss and $\mathcal{L}_{\text{RoI}}$ involves a class-conditional assigner and binary classification specified as above. Furthermore, as we are training with both datasets simultaneously, we apply the stop gradient operator on $f_{\text{gen}}(x), x \in \mathcal{D}_{\text{tgt}}$ to prevent gradient leakage from the target dataset to the box generator. During inference, we replace the box generator with the source labels, including bounding boxes and class labels $b, c$. The output consists of the transferred bounding boxes $\hat{b}$ and their validity scores $\hat{s}$. We construct the label-transferred source datasets via thresholding the validity scores: $\mathcal{D}_{\text{transferred-src}} := \{(x, \hat{b}, c)|\hat{s} \geq \sigma_c\}$ where $\sigma_c$ is a classwise threshold.

**Extensions.** The proposed LGPL can be easily extended to more specialized settings. When the two image domains are drastically different, *e.g.*, synthetic images versus real-world images, we apply an

additional $\alpha$-weighted S-CycConf (Prabhu et al., 2023) loss to Eq. 1 to align the instance features from two domains. When there are multiple source datasets, we train a LGPL for each source and target pair and apply transfer separately.

## 5 EXPERIMENTS

We detail the experiment setups and quantitatively analyze the level of annotation mismatches in Sec. 5.1. We describe the five baselines and two evaluation metrics in Sec. 5.2. We provide the main experimental results in Sec. 5.3 and find that LGPL is the *only method that consistently addresses the annotation mismatches across four scenarios and three downstream detectors*. We further find that LGPL outperforms off-the-shelf supervised domain adaptation techniques.

### 5.1 DATASETS

We create four scenarios from five real-world datasets: Cityscapes (Cordts et al., 2016), Mapillary Vistas Dataset (MVD) (Neuhold et al., 2017), Waymo (Sun et al., 2020), nuScenes, and nuImages (Caesar et al., 2020); and two synthetic datasets: Synscapes (Wrenninge & Unger, 2018) and Internal-Dataset, an internal dataset that we leave blinded for anonymity. If two datasets have different class label sets, we take the common classes. Below, we detail the four scenarios:

**1) nuScenes → nuImages (10):** This contains 'cross-modality' mismatches as the 2-D bounding boxes from nuScenes are obtained by converting the 3-D bounding boxes, creating oversized, occluded, or highly-truncated boxes on nuScenes. We sub-sample 16k images from nuImages.

**2) Synscapes → Cityscapes (7):** This contains 'class semantics' and 'human-machine misalignment' mismatches. Cyclists and bicycles have separate labels in Cityscapes, but the 'cyclist' label in Synscapes contains both. Further, Synscapes bounding boxes are generated programmatically and include highly occluded or truncated objects. For label transfer, we exclude the 'train' class in this scenario since it is too sparse in Cityscapes.

**3) Internal-Dataset → nuImages† (3):** This contains 'human-machine misalignment' mismatches, as bounding boxes in Internal-Dataset are generated programmatically and include many highly occluded or truncated objects. We take three common classes between the datasets.

**4) MVD-🚲 + nuImages-🚲 → Waymo-🚲 (1):** This contains 'class semantics' and 'annotation instructions' mismatches for the 'cyclist' class 🚲. Motorcyclists, bicyclists, and bicycles have separate labels in MVD, but they are all considered 'cyclist' in Waymo and nuImages. Further, as opposed to nuImages and MVD, Waymo ignores bicycles not on the road (See Fig. 2). To align the class label space, we treat motorcyclists, bicyclists, and bicycles as the 'cyclist' class.

We quantify the level of annotation mismatches by re-annotating a subset of nuScenes, MVD-🚲, and nuImages-🚲 according to the target annotation protocols and refer them as "gold transferred labels". TIDE analysis (Bolya et al., 2020) highlights that every scenario presents a drastically different label transfer problem, urging a general-purpose label transfer model. We detail the labeling procedure of gold transferred labels and the full TIDE analysis in Appendix A.

### 5.2 BASELINES AND EVALUATION METRICS

**Baselines.** To validate the effectiveness of LGPL, we consider two training-free transfer policies: No transfer and Statistical Normalization (SN), and two learning-based transfer policies: Pseudo-labeling (PL) and Pseudo-labeling & Noise-filtering (PL & NF). We also explore adopting image segmentation foundation model, SAM (Kirillov et al., 2023), for label transfer.

**1) No transfer:** This uses the pre-existing source labels.

**2) Statistical normalization (SN):** Wang et al. (2020) demonstrate the efficacy of rescaling bounding boxes in LiDAR-based 3-D object detection. We re-purpose their SN to 2-D detection by scaling source dataset bounding boxes to match the mean height and width of the target dataset.

**3) Pseudo-labeling (PL):** Lee (2013) creates pseudolabels for the source dataset using a detector first trained on only the target dataset. The pseudolabels are refined with standard techniques such as non-maximum suppression and then used to augment the target dataset.

| | Label transfer model | YOLOv3 | Def-DETR | Faster-RCNN |
|---|---|---|---|---|
| Source = nuScenes
Target = nuImages | No transfer | 31.24 | 39.65 | 41.25 |
| | SN | 31.95 | 39.59 | 40.79 |
| | PL | 28.67 | 39.12 | 40.49 |
| | PL & NF | 33.26 | 40.97 | 40.68 |
| | LGPL (Ours) | **34.8** +3.56 | **41.52** +1.87 | **42.6** +1.35 |
| Source = Synscapes
Target = Cityscapes | No transfer | 26.87 | 32.93 | 38.74 |
| | SN | 25.53 | 32.7 | 36.91 |
| | PL | 28.86 | 30.67 | 37.88 |
| | PL & NF | 28.27 | 33.04 | 39.05 |
| | LGPL (Ours) | **29.29** +2.42 | **34.45** +1.58 | **39.71** +0.97 |
| Source = Internal-Dataset
Target = nuImages† | No transfer | 39.17 | 46.79 | 47.91 |
| | SN | 39.07 | 47.05 | 48.05 |
| | PL | 37.87 | 47.41 | 48.5 |
| | PL & NF | 39.85 | 47.67 | 48.2 |
| | LGPL (Ours) | **41.17** +2 | **48.4** +1.61 | **48.89** +0.98 |

Table 1: **Downstream-mAP of detectors trained with transferred labels.** LGPL outperforms all baselines on all scenarios and architectures. Surprisingly, most baselines consistently fail to outperform 'No transfer' and LGPL is the *only* approach that consistently beats 'No transfer'. We use small font to denote the mAP difference versus 'No transfer', red color to indicate methods that are worse than 'No transfer', and bold the best performing label transfer models.

| | Label transfer model | YOLOv3 | Def-DETR | Faster-RCNN |
|---|---|---|---|---|
| Source = MVD-🚲 +
nuImages-🚲
Target = Waymo-🚲 | No transfer | 22.21 | 26.12 | 31.14 |
| | SN | 20.55 | 24.86 | 30.12 |
| | PL | 22.22 | 23.05 | 29.58 |
| | PL & NF | 23.78 | 27.69 | 30.69 |
| | LGPL (Ours) | **25.09** +2.88 | **27.86** +1.74 | **32.74** +1.61 |

Table 2: **Multi-source label transfer.** When multiple source datasets are presented, LGPL outperforms all baselines on all architectures as well.

**4) Pseudo-labeling & Noise-filtering (PL & NF):** This baseline improves PL by using the pre-existing source labels to filter out noisy RPN proposals by the pseudo-labeler. Inspired by Mao et al. (2020), we remove any RPN proposals that have small IoUs ($\leq 0.5$) with the source labels.

**5) SAM-transfer model:** SAM (Kirillov et al., 2023) is a general-purpose segmentation foundation model, generating accurate class-agnostic segmentation masks from prompts. We adopt SAM for label transfer by prompting it with source bounding boxes, keeping the class labels intact, and taking the bounding boxes induced by the predicted segmentation mask as the transferred bounding boxes. Since SAM-transfer model fixes only the localization errors, we consider nuScenes → nuImages for SAM-transfer model. The rest of the scenarios require determining the label validity under target annotation protocols, which is challenging for SAM-transfer model.

**Evaluation metrics.** We consider two metrics to evaluate the performance of a label transfer model:

**1) Downstream-mAP:** Training a detector with a source dataset that matches the target dataset's annotation protocol should perform better than training with a dataset that does not. To evaluate a label transfer model, we first train a downstream detector on a combined label-transferred source and target dataset to evaluate a label transfer model. Then, we use the performance of the downstream detector as the evaluate metric, denoted as downstream-mAP. Training can be augmented by domain adaptation algorithms if needed. For robustness, we validate on three popular detector architectures: YOLOv3 (Redmon & Farhadi, 2018), Deformable DETR (Def-DETR) (Zhu et al., 2021), and Faster-RCNN (Ren et al., 2015). For every experiment, we run three times except for Def-DETR due to the long training time.

**2) Transfer-mAP:** Given access to gold transferred labels for the source dataset that match the target protocol, we can directly measure the mAP of a label transfer model's output labels w.r.t. the gold transferred labels. Interestingly, we will show that although this metric bypasses the need to train a downstream detector in evaluation, it may not correlate well with the downstream performance.

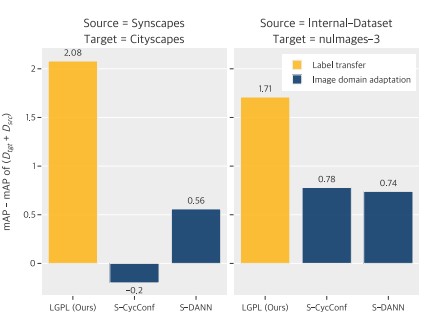

| | Label transfer model | YOLOv3 | Def-DETR | Faster-RCNN |
|---|---|---|---|---|
| | No transfer | 28.03 | 40.6 | 43.13 |
| | SN | 29.16 | 40.7 | 42.36 |
| Source = nuScenes | PL | 27.06 | 40.5 | 42.56 |
| Target = nuImages | PL & NF | 32.36 | 43.2 | 42.73 |
| | LGPL (Ours) | **34.23** +6.2 | **43.4** +2.8 | **44.63** +1.5 |
| | No transfer | 22.03 | 26.2 | 34.86 |
| Source = MVD-🚲 + | SN | 19.03 | 24.6 | 32.93 |
| nuImages-🚲 | PL | 22.43 | 23.1 | 31.49 |
| Target = Waymo-🚲 | PL & NF | 24.46 | 28.3 | 34.2 |
| | LGPL (Ours) | **26.03** +4 | **29.4** +3.2 | **36.66** +1.8 |

Table 3: **Downstream-AP**[75] **of detectors trained with transferred labels.** We highlight the differences over 'No transfer' in smaller font and color the performance deterioration in red.

Figure 4: **Label transfer vs. image domain adaptation.** LGPL improves over image domain adaptations, S-DANN and S-CycConf (Prabhu et al., 2023), by $1.6$ and $1.24$ downstream-mAP on average.

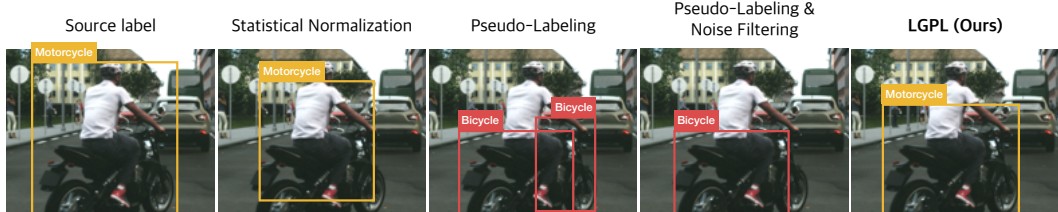

Source label | Statistical Normalization | Pseudo-Labeling | Pseudo-Labeling & Noise Filtering | **LGPL (Ours)**

Figure 5: **Synscapes → Cityscapes label transfer qualitative examples.** Synscapes annotates motorcycle as the cyclists and the motorbikes, while Cityscapes only annotates the motorbikes. Unlike LGPL, the baseline label transfer model at times hallucinate or fail to produce accurate bounding boxes. More qualitative results are in Appendix B.

**Implementation details.** We adopt same architectures for PL, PL & NF, and LGPL. Class-wise thresholds $\sigma_c$ are determined by digitizing confidence scores using the classic binning strategy (Sturges, 1926). Annotations falling into the last bin or with confidence score lower than $0.001$ are dropped. More details are provided in Appendices C and D.

## 5.3 MAIN RESULTS

▷ **LGPL outperforms all other baseline methods for every architecture.** Table 1 and Table 2 summarize our main results on downstream-mAP. Further, LGPL is the *only approach that consistently beats the 'No transfer' baseline*. On average, LGPL outperforms 'No transfer' on YOLOv3 by $2.7$, Def-DETR by $1.68$, and Faster-RCNN by $1.22$ downstream-mAP. In contrast, each of the other baselines will sometimes be worse than simply not pre-processing labels.

▷ **Transferring labels leads to higher-quality object detectors.** Our TIDE analysis (Bolya et al., 2020) of gold transferred labels reveals that annotation mismatches are almost exclusively either due to Localization or Background errors on the bounding boxes (see Appendix A.3). Intuitively, label transfer should be most effective for higher IoU thresholds where these errors are more prevalent. In Table 3, we report AP[75] for the downstream detector (see Appendix B for the full table). LGPL remains the only label transfer model that consistently beats the 'No transfer' baseline by $4.14$ for YOLOv3, $2.65$ for Def-DETR, and $1.15$ for Faster-RCNN.

▷ **LGPL outperforms off-the-shelf supervised domain adaptation.** Domain adaptation is a standard solution for training a model when there is a mismatch between a source and target data distribution. We compare LGPL and two supervised domain adaptation (SDA) approaches S-DANN and S-CycConf from Prabhu et al. (2023). See details of S-DANN and S-CycConf in Appendix C.1. Both S-DANN and S-CycConf leverage the source and the target labels to align instance features. Fig. 4 reports the relative gain versus the naïve 'No transfer' baseline on two scenarios consisting

Table 4: **Transfer-mAP and downstream-mAP.** We find that these two metrics do no strongly correlate with Spearman correlation coefficient $R_S = 0.6$. ($p$-value $> 0.1$). The ultimate goal of a label transfer model is to enhance the performance of object detectors; thus, we recommend that future work prioritizes downstream performance.

| | Label transfer model | Transfer-mAP | | Average Downstream-mAP |
|---|---|---|---|---|
| Source = nuScenes Target = nuImages | No transfer | 24.4 | | 37.38 |
| | SN | 6.5 | | 37.44 |
| | PL | 37.6 | | 36.09 |
| | PL & NF | 42.8 | | 38.03 |
| | LGPL (Ours) | **44.4** +20 | | **39.64** +2.25 |
| | | MVD-🚲 | nuImages-🚲 | |
| Source = MVD-🚲 + nuImages-🚲 Target = Waymo-🚲 | No transfer | 11.1 | 24.3 | 26.49 |
| | SN | 8 | 11.6 | 25.19 |
| | PL | 25.2 | 52.5 | 24.95 |
| | PL & NF | **30.8** +19.7 | 55.7 | 27.38 |
| | LGPL (Ours) | 25.8 | **63** +38.7 | **28.56** +2.07 |

of both real-world and synthetic images. LGPL outperforms both domain adaptation approaches, showing that closing annotation mismatches can be more effective than aligning image features.

▷ **Off-the-shelf segmentation foundation models fall short in label transfer.** Table 5 demonstrates that SAM-transfer models across different image backbones (`vit-b`, `vit-l`, and `vit-h`) all hit the wall around 38 mAP, showing that a target-specific label transfer model, *e.g.* LGPL, outperforms a general-purpose segmentation foundation model for label transfer. We observe that overly large bounding boxes confuse SAM, making it less effective in identifying the objects of interest and consequently reducing transfer quality.

| | Label transfer model | Faster-RCNN |
|---|---|---|
| Source = nuScenes Target = nuImages | No transfer | 41.25 |
| | SAM-transfer-b | 38.04 |
| | SAM-transfer-l | 38.09 |
| | SAM-transfer-h | 38.29 |
| | LGPL (Ours) | **42.6** |

Table 5: **Downstream-mAP of SAM-adopted label transfer models.**

▷ **Qualitative analysis reveals the nature of errors.** Fig. 5 visualizes examples of transferred annotations from each method in Synscapes → Cityscapes. We find that SN partially rescales the motorcycle, but PL mislabels the object and hallucinates a second bicycle. Via noise filtering, PL & NF removes the second bicycle label, but still cannot address the class label mismatch. We show more qualitative results in Appendix B and find that LGPL learns to correctly transfer the source bounding boxes, drop the occluded ones, and preserve the correct labels.

▷ **Measuring downstream-mAP is necessary for evaluating label transfer models.** Table 4 reports transfer-mAP, which measures the mAP between the predicted transferred labels and the gold transferred labels. We find that high improvements in transferred-mAP map to only minor improvements in downstream-mAP, and moreover, the highest transferred-mAP does not always map to the highest downstream-mAP. The Spearman Rank Correlation (Spearman, 1987) coefficient between these scores is $R_S = 0.6$ ($p$-value $> 0.1$), suggesting that is no significant correlation between the two metrics. We conclude that visually analyzing the transferred labels is insufficient to determine whether label transfer is effective, and we must train the downstream model to verify.

## 6 CONCLUSION

We introduce label transfer for object detection, which rectifies annotation mismatches by adapting class labels and bounding boxes to the target annotation protocol. Our method, Label-Guided Pseudo-Labeling (LGPL), adopts two-stage object detectors with careful design, allowing easy extensibility. Once trained, LGPL can seamlessly integrate into existing training workflows, enhancing downstream detection systems across various architectures and training objectives. Results consistently showcase consistent improvements over the original source labels, yielding gains of $1.88$ in downstream-mAP and $2.65$ in downstream-AP$^{75}$. Additionally, our experiments demonstrate LGPL's superiority over off-the-shelf supervised domain adaptation techniques, which focus solely on aligning instance features, in effectively improving downstream object detectors by addressing annotation mismatches.

**Limitations.** Our proposed label transfer problem assumes that we do not need to transfer class labels or generate new bounding boxes not provided by the source dataset. In practice, systematic class label transfer may be performed by alternate pre-processing. We envision future work to explore jointly correcting class label mismatches and bounding boxes.

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

We detail the dataset statistics and additional analysis of each scenario in Section A. Then, we provide full experimental results and qualitative results in Section B. We evaluate the label-transferred source datasets by the downstream performances. We provide additional details of the downstream detector in Section C. Lastly, we provide implementation details in Section D for reproducibility.

# A DATASET ANALYSIS

## A.1 DATASET DETAILS

In this section, we describe the details of the seven datasets and the dataset statistics in Table 6:

- Cityscapes Cordts et al. (2016) contains 2975 training and 500 validation images. The dataset provides fine pixel-level annotations labeled by in-house annotators. We obtain the pixel-based bounding boxes from the semantic segmentation masks.
- Mapillary Vistas Dataset (MVD) Neuhold et al. (2017) contains 18,000 images with pixel-accurate and instance-specific human annotations over 124 categories. We obtain the pixel-based bounding boxes from the semantic segmentation mask
- Waymo Open dataset (Waymo) Sun et al. (2020) contains 1950 segments in diverse geographies and conditions and exhaustively annotated with 2-D and 3-D bounding boxes using production-level labeling tools . We sub-sample every 10th frame as our dataset, ending up with around 80,000 images.
- nuScenes Caesar et al. (2020) contains 1000 driving scenes in Boston and Singapore. Around 84,000 keyframes are labeled with 3-D bounding boxes labeled by in-house annotators. We obtain the 2-D bounding boxes by projecting the 3-D bounding boxes to the 2-D plane
- nuImages Caesar et al. (2020) is a stand-alone large-scale image dataset with 67,000 images. The dataset provides 2-D bounding boxes labeled by in-house annotators.
- Synscapes Wrenninge & Unger (2018) is a photorealistic synthetic dataset that is designed to be structurally similar to Cityscapes Cordts et al. (2016), containing 25,000 images. We use the 2-D bounding provided in the dataset.
- Internal-Dataset is a photorealistic private synthetic dataset of 48,000 driving scenes.

## A.2 SCENARIO DETAILS

In this section, we describe the details of how we use them to construct the four object detection scenarios:

- nuScenes → nuImages: This scenario presents 'cross-modality labels' annotation mismatch. We obtain the 2-D bounding boxes from existing 3-D bounding boxes in nuScenes, which leads to oversize, occluded, or highly-truncated boxes or boxes. Both datasets are collected with the same sensor and similar annotation instructions, leading to minimum annotation mismatches in 'class semantics' and 'annotation instructions'. We use all ten classes in nuImages including car, truck, trailer, bus, construction vehicle, bicycle, motorcycle, pedestrian, traffic cone, and barrier.
- Synscapes → Cityscapes: This scenario presents 'class semantics' and 'human-machine misalignment' annotation mismatches. Since there is no class rider in Synscapes, we exclude it in this scenario, resulting in a seven-way object detection problem. Additionally, all label transfer model do not transfer the class 'train' since there are only 168 'train' instances, accounting for 0.3% of all Cityscapes annotations. Cityscapes labels 'cyclist', 'motorcycle', and 'bicycle' into three separate classes, whereas the 'motorcycle' and 'bicycle' in Synscapes include riders. The seven classes include person, car, truck, bus, train, motorcycle, and bicycle.
- Internal-Dataset → nuImages†: The scenario presents 'human-machine misalignment' annotation mismatch. We take three common classes from both datasets, including car, bicycle, and pedestrian.

| | #classes | #source images | #instances per source image | #target images | #instances per target image |
|---|---|---|---|---|---|
| nuScenes → nuImages | 10 | 84390 | 6.85 | 15591 | 8.02 |
| Synscapes → Cityscapes | 7 | 25000 | 46.17 | 2975 | 18.17 |
| Internal-Dataset → nuImages† | 3 | 47994 | 7.33 | 80291 | 5.14 |
| MVD-🚲 + nuImages-🚲 → Waymo-🚲 | 1 | 8622 | 1.7 | 3649 | 1.41 |

Table 6: **Datasets statistics.**

| | Waymo | nuImages | nuScenes | Cityscapes | MVD | Synscapes |
|---|---|---|---|---|---|---|
| Waymo | - | CS, AI | CS, AI, CM | CS, CM | CS, CM | CS, HMM |
| nuImages | - | - | CM | CS, CM | CS, CM | CS, HMM |
| nuScenes | - | - | - | CS, CM | CS, CM | CS, HMM, CM |
| Cityscapes | - | - | - | - | - | CS, HMM, CM |
| MVD | - | - | - | - | - | CS, HMM, CM |
| Synscapes | - | - | - | - | - | - |

Table 7: **Annotation mismatches of each pair of object detection datasets.** Almost every pair of datasets has annotation biases, and among all types of annotation biases and CS is the most common one. Notice that we do not aim to exhaustively find out all the annotation mismatches. Instead, we describe the most obvious ones in this table. CS: Class semantics; AI: Annotation instructions; HMM: Human-machine misalignment; CM: Cross-modality labels.

- MVD-🚲 + nuImages-🚲 → Waymo-🚲: This scenario presents 'class semantics' and 'annotation instructions' annotation mismatches. We take the cyclist class from each dataset and subsample the images accordingly. MVD has two separate classes 'bicyclist' and 'other rider', which we map to a superclass 'cyclist'. The class 'bicycle' in nuImages consists of all vision bikes in the scenes and includes all the riders and passengers, if any, on the bikes. On the other hand, the class 'cyclist' exclude bikes parked on sidewalks.

We pinpoint four different types of annotation mismatches from our comprehensive survey in Section 3. In Table 7, we provide our comprehensive study on the annotation biases of each pair of datasets.

### A.3 Quantifying annotation mismatches with gold transferred labels

In section 3, we propose the taxonomy for common annotation mismatches. To understand the annotation mismatches quantitatively , we analyze two scenarios: nuScenes → nuImages and MVD-🚲 +nuImages-🚲 → Waymo-🚲. We manually annotated nuScenes, MVD-🚲, nuImages-🚲 according to their target annotation protocols and refer them as gold transferred labels. Specifically, we first draw all the bounding boxes on the images and ask the expert annotators to resize or remove the bounding boxes. For example, the annotators need to remove the bicycle labels if they are parked on the sidewalk in nuImages-🚲.

With the gold transferred labels, we perform TIDE analysis (Bolya et al., 2020) to dissect the gaps presented between the pre-existing labels and the gold transferred labels. Note that since both set of labels are carefully annotated by human, error types like duplication, classification, etc. are close to zero. Table 8 shows that nuScenes → nuImages has a high localization error of 11.1, while nuImages-🚲 → Waymo-🚲 has more hallucinated background labels with background error of 54.24. MVD-🚲 → Waymo-🚲 suffers from high localization and background errors of 14.26 and 30.32, respectively. The TIDE analysis highlights that every scenario presents a drastically different label transfer problem, urging a general-purpose label transfer model.

The taxonomy relates the annotation mismtaches with the difference in bounding boxes. For example, Table 7 and the TIDE analysis in Table 9 together reveal that the cross-modality (CM) mismatch in nuScenes nuImages leads to severe bounding box shift, but relatively little over-detection issues.

|  | #images | #transferred instances per image | #source instances per image |
|---|---|---|---|
| nuScenes | 114 | 4.95 | 6.85 |
| MVD-🚲 | 125 | 0.87 | 2.11 |
| nuImages-🚲 | 150 | 0.14 | 1.42 |

Table 8: **Statistics of gold transferred labels.**

|  | Source = nuScenes Target = nuImages | Source = MVD-🚲 Target = Waymo-🚲 | Source = nuImages-🚲 Target = Waymo-🚲 |
|---|---|---|---|
| Localization | 11.1 | 14.26 | 0. |
| Background | 0.83 | 30.32 | 54.24 |

Table 9: **TIDE analysis of the pre-existing source labels *w.r.t.* gold transferred labels.** TIDE Bolya et al. (2020) analyzes the source of errors between two set of detection labels. The higher the number is, the more prevalent the type of errors exist between labels. Other errors types are close to zero.

# B ADDITIONAL EXPERIMENTAL RESULTS

We provides the full table of Table 1 in Table 10. The standard deviation across three runs shown in smaller font. We provide additional 'Target only' results as the reference numbers. Note that for 'Target only', we train the detector only on the target dataset. We exclude Deformable DETR Zhu et al. (2021) since it requires longer training compared to other two detectors. We observe that the standar deviations of LGPL are no more than 0.33, which validates the robustness of LGPL's improvements.

We provide the full table of Table 3 in Table 11. We provide additional 'Target only' results as the reference numbers. Note that for 'Target only', we train the detector only on the target dataset. Similar to our conclusion in Section 5.3, LGPL brings greater improvements when evaluated with higher IoU, *i.e.* downstream-AP$^{75}$.

Aside from the above quantitative results, we provide additional qualitative results in Fig. 6, Fig. 7, Fig. 8, and Fig. 9

# C DOWNSTREAM EVALUATION

For Faster-RCNN Ren et al. (2015) and Deformable DETR Zhu et al. (2021), we adopt ResNet-50 He et al. (2016) as the image backbone For YOLOv3 Redmon & Farhadi (2018), we adopt DarkNet-53 as the image backbone. We sweep the learning rate for each downstream detector with grid search, while other hyper-parameters remain unchanged from the origin papers. When the source and the target image domains are drastically different, Synscapes $\rightarrow$ Cityscapes and Internal-Dataset $\rightarrow$ nuImages†, we apply image domain adaptation and use $\alpha_{\text{image}}$ to balance the task loss and the image domain adaptation loss. For each experiment, we run on three random seeds, except for Deformable DETR due to its long training time. We describe the range of the hyper-parameters sweep in Table 12.

For Faster-RCNN and Deformable DETR, we resize the image to $(1800, 900)$ and randomly flip the image horizontally in the Synscapes $\rightarrow$ Cityscapes scenario, and resize the image to $(1600, 900)$ and randomly flip the image horizontally in the other three scenarios. For YOLOv3, we resize the image to $(1500, 800)$ and randomly flip the image horizontally in all scenarios.

## C.1 SUPERVISED DOMAIN ADAPTATION (PRABHU ET AL., 2023)

Typical domain adaptation approaches assume the access to a large amount of labelled data from the source distribution and unlabeled or sparsely labelled data from the target distribution. In contrast,

| | Label transfer model | YOLOv3 | Faster-RCNN |
|---|---|---|---|
| Source = nuScenes
Target = nuImages | Target only | 27.72 ± 0.05 | 30.85 ± 0.24 |
| | No transfer | 31.24 ± 0.48 | 41.25 ± 0.13 |
| | SN | 31.95 ± 0.18 | 40.79 ± 0.1 |
| | PL | 28.67 ± 0.31 | 40.49 ± 0.05 |
| | PL & NF | 33.26 ± 0.02 | 40.68 ± 0.27 |
| | LGPL (Ours) | **34.8** ± **0.13** | **42.6** ± **0.1** |
| Source = Synscapes
Target = Cityscapes | Target only | 19.46 ± 0.23 | 25.76 ± 0.46 |
| | No transfer | 26.87 ± 0.67 | 38.74 ± 0.15 |
| | SN | 25.53 ± 0.19 | 36.91 ± 0.43 |
| | PL | 28.86 ± 0.57 | 37.88 ± 0.32 |
| | PL & NF | 28.27 ± 0.38 | 39.05 ± 0.2 |
| | LGPL (Ours) | **29.29** ± **0.17** | **39.71** ± **0.16** |
| Source = Internal-Dataset
Target = nuImages† | Target only | 35.25 ± 0.23 | 40.14 ± 0.33 |
| | No transfer | 39.17 ± 0.13 | 47.91 ± 0.14 |
| | SN | 39.07 ± 0.23 | 47.99 ± 0.09 |
| | PL | 37.87 ± 0.07 | 48.59 ± 0.08 |
| | PL & NF | 39.85 ± 0.09 | 48.19 ± 0.12 |
| | LGPL (Ours) | **41.17** ± **0.18** | **48.89** ± **0.11** |
| Source = MVD-🚲 +
nuImages-🚲
Target = Waymo-🚲 | Target only | 19.27 ± 0.2 | 23.64 ± 0.25 |
| | No transfer | 22.21 ± 0.38 | 31.14 ± 0.12 |
| | SN | 20.55 ± 0.43 | 30.12 ± 0.11 |
| | PL | 22.22 ± 0.22 | 29.58 ± 0.46 |
| | PL & NF | 23.78 ± 0.92 | 30.69 ± 0.5 |
| | LGPL (Ours) | **25.09** ± **0.23** | **32.74** ± **0.33** |

Table 10: **Downstream-mAP of detectors trained with transferred labels.** LGPL outperforms all baselines on all scenarios and architectures. Further, it is the only approach that consistently beats 'No transfer'. We use small font to denote the standard deviation over three runs, red color to indicate methods that are worse than 'No transfer', and bold the best performing label transfer models.

| | Label transfer model | YOLOv3 | Def-DETR | Faster-RCNN |
|---|---|---|---|---|
| Source = nuScenes Target = nuImages | Target only | 25.53 | 38 | 28.16 |
| | No transfer | 28.03 | 40.6 | 43.13 |
| | SN | 29.16 | 40.7 | 42.36 |
| | PL | 27.06 | 40.5 | 42.56 |
| | PL & NF | 32.36 | 43.2 | 42.73 |
| | LGPL (Ours) | **34.23** +6.2 | **43.4** +2.8 | **44.63** +1.5 |
| Source = Synscapes Target = Cityscapes | Target only | 17.56 | 27.5 | 21.76 |
| | No transfer | 26.3 | 33 | 40.4 |
| | SN | 23.56 | 33.6 | 38.66 |
| | PL | 27.53 | 29.1 | 39.3 |
| | PL & NF | 28.1 | 32.06 | **40.9** +0.5 |
| | LGPL (Ours) | **29.06** +2.76 | **35.1** +2.1 | 40.7 |
| Source = Internal-Dataset Target = nuImages† | Target only | 34.4 | 43.8 | 39.3 |
| | No transfer | 37.43 | 49.4 | 51 |
| | SN | 36.93 | 49.7 | 51.67 |
| | PL | 35.53 | 50.6 | 51.93 |
| | PL & NF | 39.1 | 50.7 | 51.06 |
| | LGPL (Ours) | **41.06** +3.63 | **51.9** +2.5 | **52** +1 |
| Source = MVD-🚲 + nuImages-🚲 Target = Waymo-🚲 | Target only | 19.36 | 18.6 | 24.86 |
| | No transfer | 22.03 | 26.2 | 34.86 |
| | SN | 19.03 | 24.6 | 32.93 |
| | PL | 22.43 | 23.1 | 31.49 |
| | PL & NF | 24.46 | 28.3 | 34.2 |
| | LGPL (Ours) | **26.03** +4 | **29.4** +3.2 | **36.66** +1.8 |

Table 11: **Downstream-AP$^{75}$ of detectors trained with tranferred labels.** We highlight the differences over 'No transfer' in smaller font and color the performance deterioration in red.

| | Hyper-parameter name | Sweep range |
|---|---|---|
| YOLOv3 | Learning rate | $0.0001, 0.0002, 0.0003$ |
| | $\alpha_{\text{image}}$ | $0.1, 0.2, 0.3$ |
| Deformable DETR | Learning rate | $0.001, 0.002, 0.003$ |
| | $\alpha_{\text{image}}$ | $0.1, 0.2, 0.3$ |
| Faster-RCNN | Learning rate | $0.04, 0.05, 0.06$ |
| | $\alpha_{\text{image}}$ | $0.1, 0.2, 0.3$ |

Table 12: **The range of the hyper-parameters sweep for different downstream detectors**

CARE Prabhu et al. (2023) formulates *supervised domain adaptation* problem, where the target labelled data is accessible. This scenario is important in high-stakes industrial applications where the annotation costs decrease due to its scale.

CARE proposes several ways to align the instance features for supervised domain adaptive object detectors by explicitly leveraging the source and the target labels and extending several popular unsupervised domain adaptation methods to their supervised version. We take S-DANN and S-CycConf from CARE. Additionally, in contrast to CARE (Prabhu et al., 2023), we empirically find that cycle confusion loss works slightly better than cycle consistency loss (Dwibedi et al., 2019) in our scenarios.

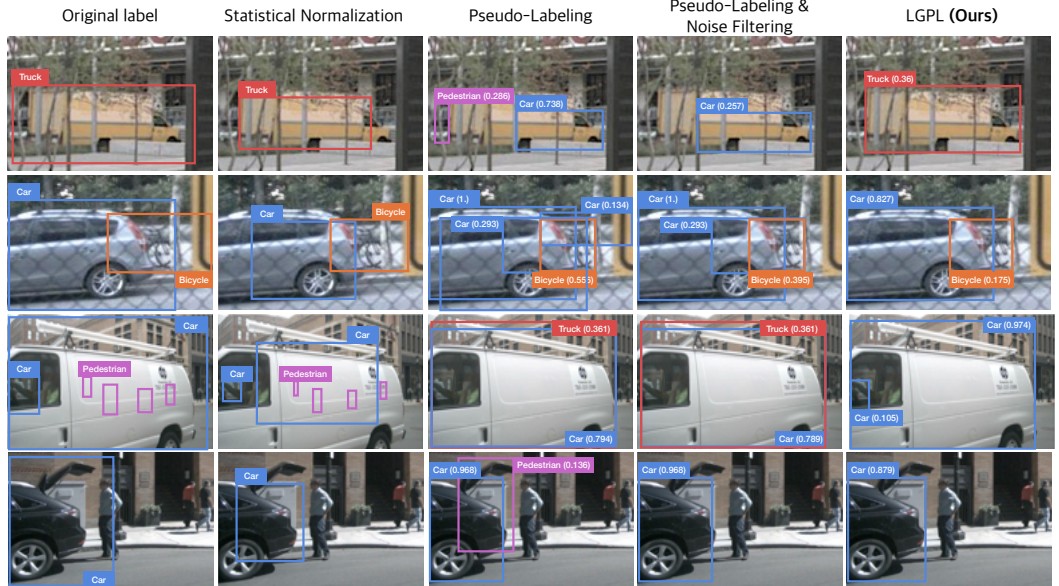

Figure 6: **Qualitative results of nuScenes → nuImages.** The top three examples show that LGPL successfully transfers the original overly large bounding boxes and removes the fully-occluded objects. The last row shows that all approach fails to annotate the car with opened trunk, potential due to the lack of similar examples in the training phase. The scores in the bracket represent the validity in the target dataset.

## D    IMPLEMENTATION DETAILS OF LABEL TRANSFER MODELS

All data-driven label transfer models (PL, PL & NF, and LGPL) adopt Cascade-RCNN Cai & Vasconcelos (2018) and use ImageNet-pretrained HRNet-w32 Sun et al. (2019) as the image backbone with batch size 16. Cascade-RCNN extends Faster-RCNN Ren et al. (2015) with a multi-stage RoI head. Following prior work, the IoU thresholds are set as 0.5, 0.6, and 0.7 at each stage. We add CycConf loss Wang et al. (2021) with weight $\alpha$ in Section 4.3 to align image features in Synscapes → Cityscapes and Internal-Dataset → nuImages†. The learning rate and $\alpha$ are tuned with grid search, while other hyper-parameters remain unchanged from the original Cascade-RCNN. We sweep the learning rate with the values $0.01, 0.02, 0.03, 0.04$ and $\alpha$ with the values $0.01, 0.02, 0.03$. All experiments are run on NVIDIA Tesla V100 GPUs.

### D.1    CHOICE OF CLASS-CONDITIONAL THRESHOLD $\sigma_c$

We first empirically found that adopting a class-agnostic threshold leads to performance worse than "No transfer". However, treating the class-conditional threshold as another set of hyperparameters makes the hyperparameter search extremely computationally expensive since every evaluation requires training a new object detector. We, therefore, choose $\sigma_c$ by digitizing the confidence score (Sturges, 1926) for each class and apply it to all the label transfer models. Annotations falling into the last bin or with a confidence score lower than $0.001$ are dropped. In this way, the classes that are more challenging get the lower thresholds and vice versa. We leave the choice of the class-conditional threshold to future research.

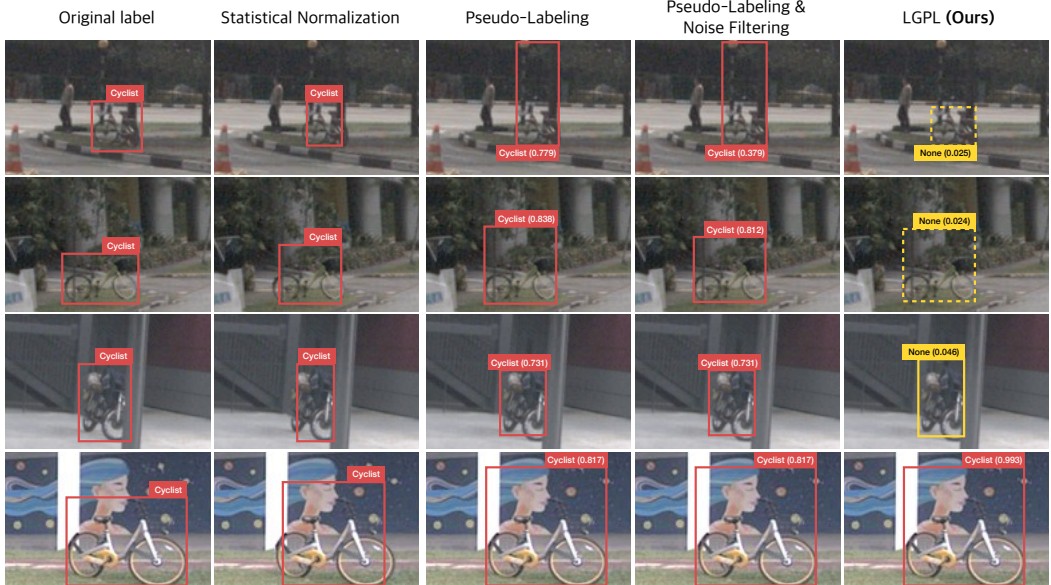

Figure 7: **Qualitative results of nuImages-🚲 → Waymo-🚲.** The top three examples show that LGPL successfully removes the bikes on the sidewalks by assigning them low confidence scores. The last row shows a bike parked in front of a painted wall. All methods fail to remove the bike label due to the painting in the background. Yellow dashed bounding boxes are removed due to the low confidence scores. The scores in the bracket represent the validity in the target dataset.

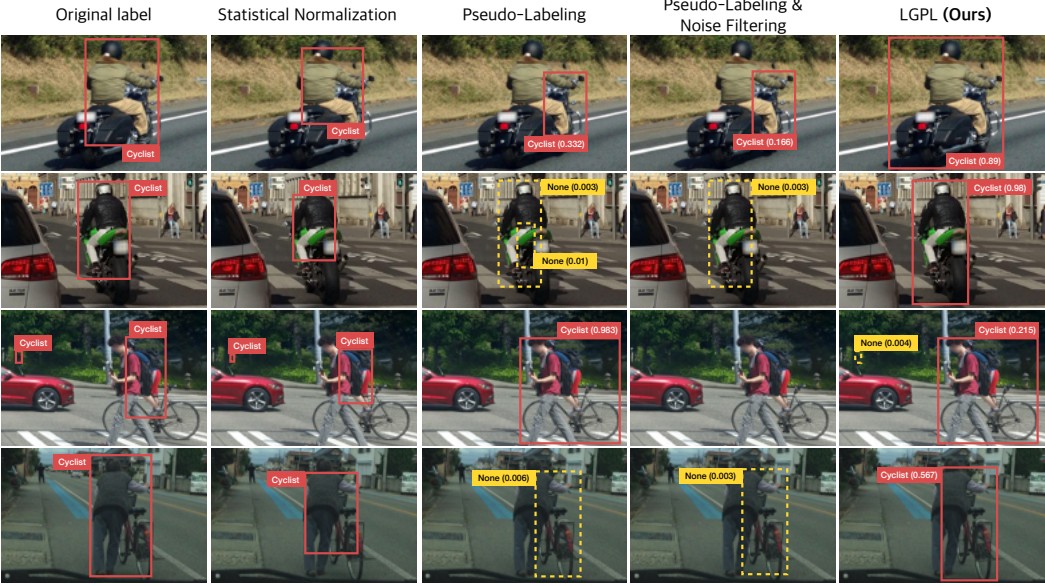

Figure 8: **Qualitative results of MVD-🚲 → Waymo-🚲.** In the top row, PL and PL & NF fail to recognize the complete motorcycle, while LGPL successfully transfers the bounding boxes to encompass both the motorcycle and the motorcyclist. In the third row, PL & NF fails to detect anything since there is no prediction that has an IoU of $\geq 0.5$ with any source bounding box. The last row shows a person walking with a bike. According to the Waymo annotation instructions, when a pedestrian is getting onto a bicycle, they are labeled as a cyclist if they are in the riding position. To the best of our knowledge, this label should be removed, and only PL and PL & NF successfully remove it. Due to its data-driven nature, LGPL encounters difficulties in these ambiguous cases. Yellow dashed bounding boxes are removed due to the low confidence scores. The scores in the bracket represent the validity in the target dataset.

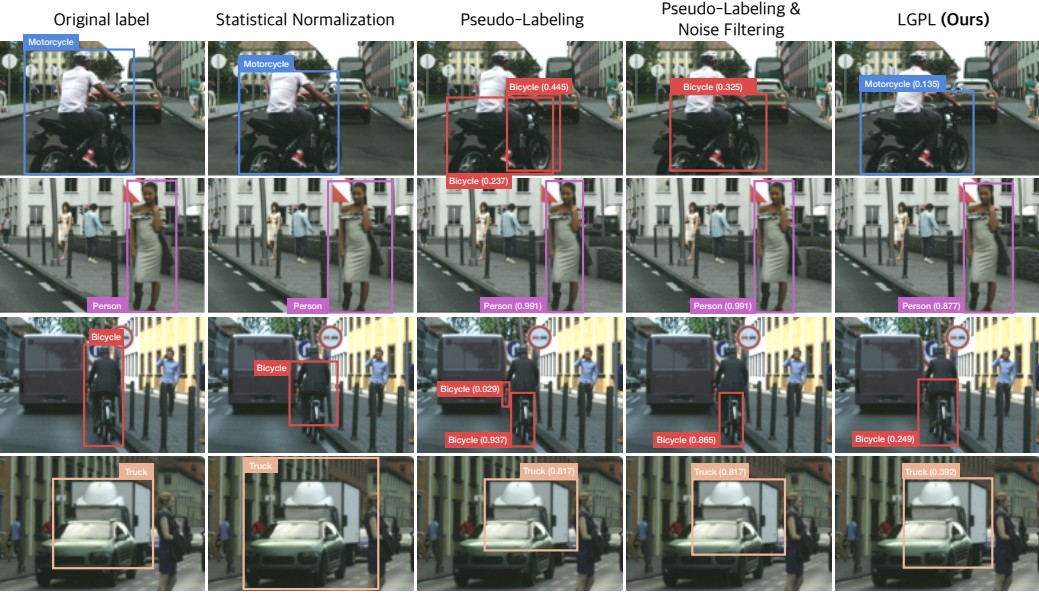

Figure 9: **Qualitative results of Synscapes → Cityscapes.** In the top row, both PL and PL & NF recognize the motorcycle as a bicycle. The bottom three rows shows that all baselines approaches fail to localize the object accurately. The scores in the bracket represent the validity in the target dataset.

