# OpenReview forum: "Transferring Labels to Solve Annotation Mismatches Across Object Detection Datasets"
_ICLR.cc/2024/Conference — ICLR 2024 poster_

### Official Review · Reviewer_2kAn · 2023-10-23

**Soundness:** 2 fair
**Presentation:** 1 poor
**Contribution:** 2 fair
**Rating:** 6
**Confidence:** 3

**Summary:**

In this paper, the authors study the problem of addressing bounding-box annotation mismatches across different datasets. For this end, they propose training a "translator" network that converts the bounding box annotations in the source dataset into the target dataset.

**Strengths:**

+ Annotation mismatch across datasets is a severe problem in OD.
+ Good results.

**Weaknesses:**

1. I had problem following the scope of the paper.

1.1. The paper introduces a very general alignment setting in Introduction + Figure 1, which is not matched with what is really performed. I had to make several loops between Method & Introduction + Figure 1 to understand what is going on.

1.2. "1) We consider scenarios where the class label spaces are matched but the annotation protocols are different, therefore, leading to annotation mismatches (see Section 3). 2) We optimize the target performances, while multi-dataset detection optimizes the average performances of all datasets considered." => The authors should provide convincing arguments + results on why this narrowed-down scope of the mismatch problem is significant.

1.3. "In this work, we assume that (1) the class labels are the same between the source and the target labeling functions and (2) the source labeling function either detects or over-detects all the objects specified in the target labeling function." => Again, the validity of these assumptions should be justified.

1.4. "We optimize the target performances, while multi-dataset detection optimizes the average performances of all datasets considered." => Why is it not better to have a multi-dataset version?

1.5. "label translation" is misleading. With the general coverage in the Introduction, I expected language translation to be performed in the method. "Recall from our main assumption that label translation does not require generating a correct class label, but only determining whether the object should be labeled as well as the corresponding bounding box." => I have difficulty calling this translation.

1.6. Section 3: It would be worthwile to extend this section with analyses. As such, it is rather weak.

2. Experimental evaluation is limited. It is not clear why YOLOv3 is chosen, compared to its newer versions or other one-stage detectors?

3. Most importantly, as bbox localization changes through the proposed method, the paper should quantify localization directly and evaluate performance with such a measure. I would recommend the localization term of the Localization Recall Precision metric.


Minor comments:
- Fig 1 left: Cyclist => rider for the MVD dataset.
- Introduction: "This work characterizes four types of annotation mismatches," => It would be nice to briefly summarize here what these four types are.
- Figure 3: The figure gives a feeling that the 'cyclist' label belongs to both datasets, which makes it difficult to understand what is being translated into what.

**After the rebuttal**

The authors have been able to partially address my concerns about the significance of the taxonomy and the experimental evaluation. For taxonomy to be considered a contribution, I expect more analyses & justifications. The experiments with a SOTA model should also be more comprehensive as YOLOv8-nano is hardly a SOTA model. However, the authors did the best of what could be done in a short amount of time and I find the extended discussion on the taxonomy and the new results promising. Therefore, I would like to increase my original recommendation.

**Questions:**

See Weaknesses.

---

> ### Author Response · Authors · 2023-11-15
> **Response to Reviewer 2kAn**
>
> $\textbf{Dear Reviewer 2kAn}$
>
> Thank you for your positive comments regarding the good results presented in the paper.
>
> In the following, we answer the reviewer's concern with the scope of the paper:
> 1. **Confusion with the introduction and Figure 1**: We introduce the label translation problem as a general and prevalent type of mismatches among object detection datasets. While resolving arbitrary label mismatches is extremely non-trivial, we consider our streamlined version of addressing specifically bounding box mismatches as the necessary first step in this novel line of research. Fortunately, our metric of concern (target performance) provides strong evidence that solving the simpler problem remains effective. We would appreciate if the reviewer could expand on any points that they feel we haven't addressed.
> 2. **Narrowing down the scope of the mismatch problem**:
>     - We argue that optimizing only the target dataset performance is not narrowing down from multi-dataset object detection. Instead, they have distinct values depending on the application. Optimizing target dataset performance is particularly important when there is a specific target domain of interest, such as in Sim2Real. The performance on simulated data is relatively unimportant and only serves the purpose of improving target performance on real-world data.
>     - While we narrowed down the mismatch problem to a shared class label space, LGPL can handle the setting when two datasets have only partially overlapped categories, by translating the overlapped categories and leaving the other as is.
> 3. **Assumptions in Section 4.1**: The assumptions are made based on the observation from seven datasets surveyed in Section 3.
>     - To further justify the second point “the source labeling function either detects or over-detects all the objects specified in the target labeling function”, we manually collect the gold translated labels and perform TIDE analysis to understand the types of translation required (See the details in Section 5.1 and Section A.3). The results in Table 9 show the low value of “Missed error” and high value of “Background error”, which supports the need to fix background errors.
> 4. **A multi-dataset version**: As pointed out in the second point (`Narrowing down the scope of the mismatch problem`), performing multi-dataset object detection has distinct application values that differ from our goals in this work.
> 5. **The term "label translation"**: We thank the reviewer for the thoughtful concerns. The confusion with language translation is unfortunate, and we are keen to hear suggestions on how this might be mitigated in future.
> 6. **The role of Section 3**: Section 3 introduces the concept of annotation mismatches and categorizes them into four common types. While it is challenging to quantitatively measure how severe annotation mismatches are between each pair of datasets, we manually collect gold translation labels and quantify the TIDE errors in Section 5.1 and Appendix A.3.
>
>
>
>
> **Empirical evaluation**
>
> The reviewer asked why we chose YOLOv3 instead of newer variants or one-stage detectors. In fact, we specifically choose three different types of object detectors: two-stage object detector (Faster-RCNN), one-stage object detector (YOLOv3), and transformer-based object detector (Def-DETR).
>
> We choose YOLOv3 out of three YOLO variants in MMdetection due to its stable training. But in any case, we feel that we have adequately demonstrated the robustness of our technique to different object detectors. If the reviewer would like to see the performance on another detector, please kindly provide us with a reference and we will aim to incorporate these results into a later version.
>
> We would also want to emphasize that we have tested on four different scenarios with three different architectures as well as ablations with model-centric alternatives such as domain adaptation and foundation model alternatives such as SAM.
>
> **Suggestions of LRP metric**
>
> We appreciate the reviewer's great suggestion of the LRP metric and we will include the LRP metric in our camera-ready version.
>
> **Minor comments**
>
> We appreciate the feedback and have revised the paper submission accordingly.

---

> > ### Comment · Reviewer_2kAn · 2023-11-16
> > **Re: Response to Reviewer 2kAn**
> >
> > Dear authors,
> >
> > Thank you for the detailed response.
> >
> > * Label translation: I would recommend "label transfer" as an alternative. "Label translation" is also inappropriate from a bbox transformation point of view in that you are not simply translating the boxes, you are performing a transformation on them.
> >
> > * If you can resolve this ambiguity, it will be easier to follow the paper, without making multiple loops between the Method and Introduction sections.
> >
> > * "See the details in Section 5.1 and Section A.3" => Please make these cross-references in places where you are stating your assumptions. Without these, a reader cannot follow the validity of your assumptions.
> >
> > * Difficulty of justifying the taxonomy: I understand that this might be difficult. As it is, without concrete analyses and justifications, it is not possible to see the extent/severeness of these problems discussed in Section 3. As such, I find the taxonomy contribution claim of the paper not substantiated.
> >
> > * YOLO v3 results: Your current results suggest that the improvements with strong detectors (Def-DETR and Faster R-CNN) are less. Therefore, one wonders how your method would have performed with a stronger/newer one-stage detector. I suspect that the improvement will be less. I don't find the use of MMdetection nor stable training a good excuse. People, including me, do use other frameworks and can train newer detectors in a stable manner.
> >
> > * I also would like to see a SOTA comparison. Def-DETR and Faster R-CNN are strong detectors, but they are not SOTA. If bbox translation does not yield SOTA results (improvements with a SOTA method), then it is not clear why it would be needed.
> >
> > Thank you,
> > Best

---

> > > ### Author Response · Authors · 2023-11-23
> > >
> > > $\textbf{Dear Reviewer 2kAn}$
> > >
> > >
> > > Thanks for the quick reply and constructive suggestions.
> > >
> > > **From label translation to label transfer**: Thanks for this suggestion. We agree this may clarify potential confusion and have revised the paper accordingly.
> > >
> > >
> > > **Scope ambiguity and in-place cross-references**: We have fixed the submission to clarify these points.
> > >
> > >
> > > **Justifying the taxonomy**: We are happy to pursue any suggestions for how we may better validate the taxonomy. Our revision clarifies the following points.
> > >
> > >
> > > The taxonomy is proposed not to exhaustively list out every possible annotation mismatch, but to provide a deeper understanding to the problem and we expect future works can build on top of the taxonomy. For example, currently in Table 7, we compare 14 dataset pairs and characterize which mismatches exist with each pair.
> > >
> > >
> > > It is challenging to measure how many examples suffer from certain types of mismatches since ideally, we would need to manually count the number/type of mismatches for each pair of datasets. However, this is beyond scope since it is comparable to relabeling the entire datasets. Instead in Table 8 and 9, we have manually relabeled 300 images from nuScenes, MVD, and nuImages according to each other's label styles. The taxonomy relates the annotation mismatches with the difference in bounding boxes. For example, Table 7 and the TIDE analysis in Table 9 together reveal that the cross-modality (CM) mismatch in nuScenes $\rightarrow$ nuImages leads to severe bounding box shift, but relatively little over-detection issues.
> > >
> > >
> > > **Stronger one-stage object detector and SOTA object detector**: We agree that applying LGPL on SOTA object detectors may potentially yield smaller gains, as it is natural to observe a ceiling effect as the base object detectors get stronger. However, we emphasize that this work is not about building a SOTA algorithm, but showing that our pre-processing can *consistently yield some gains essentially for free*.
> > >
> > >
> > > **New experiments with stronger one-stage object detector**: We have performed new experiments using [YOLOv8-nano](https://github.com/open-mmlab/mmyolo/tree/main/configs/yolov8) [1], as our evaluating object detector for nuScenes $\rightarrow$ nuImages. The findings here align with our results in Section 5.3: 1)LGPL *consistently* outperforms all the baselines out-of-the-box and 2) Translating labels via LGPL leads to higher-quality object detectors.
> > >
> > >
> > > |                | YOLOv8-nano Downstream-mAP | YOLOv8-nano Downstream-AP$^{75}$ |
> > > |----------------|----------------|----------------|
> > > | No-translation | 20.5          | 19.7               |
> > > | PL             |  16.3         | 16               |
> > > | PL \& NF       | 20.7          | 20.8               |
> > > | LGPL           | 25           | 25.2               |
> > >
> > >
> > > To finish these runs in time, we only trained on 1 random seed, reduced the image resolution to [1024, 512], and used default hyperparameters for all settings. Our original experiments had 3 seeds, a resolution of [1500, 800], and extensive hyperparameter grid searching. We plan to complete these results on YOLOv8 in the camera-ready version.
> > >
> > >
> > > [1] YOLOv8, https://github.com/ultralytics/ultralytics

---

> > > > ### Comment · Reviewer_2kAn · 2023-11-23
> > > > **Re: Official Comment by Authors**
> > > >
> > > > Dear authors,
> > > >
> > > > Thank you for the comprehensive and detailed response. Your response does partially address my concerns about the taxonomy and the evaluation with the SOTA models. You did the best of what could be done in a short amount of time, thank you. I will increase my recommendation.
> > > >
> > > > Best

---

### Official Review · Reviewer_MRBF · 2023-10-27

**Soundness:** 3 good
**Presentation:** 3 good
**Contribution:** 3 good
**Rating:** 6
**Confidence:** 4

**Summary:**

This paper attempts to tackle annotation mismatches in multi-dataset training for object detectors. It points out that different datasets may have different definitions or annotation protocols to the same category, leading to annotation mismatches. Four kinds of annotation mismatches are highlighted, i.e. Class semantics, Annotation instructions, Human-machine misalignment, Cross-modality labels. To address those mismatches, it proposes to train a label translator to translate annotations of source datasets to the target datasets. The label translator follows the design of two-stage detector (e.g. faster rcnn) with a RPN and RoI head. During the training of the translator, the RPN is trained with source datasets and generates source-style boxes. The RoI head converts the generated source-style boxes to the target-style. During inference, the RoI head converts ground-truth annotations of the source dataset (both boxes and class labels) to the target-style. Experiments shows that the proposed label translator improves various detectors on certain pairs of datasets.

**Strengths:**

1. The task of annotation mismatches for the same category is interesting. Based on my knowledge, studies of multiple dataset training seldom explore annotation issues for the same category.

2. It's interesting to leverage RoI head to covert annotations of the source datasets to the target datasets.

3. The proposed translator achieves good performance on various detectors and datasets.

4. The paper is well written and easy to follow.

**Weaknesses:**

1. The assumption is too strong that the source and the target datasets have the same categories. In multi-dataset training, it is a common case that different datasets not only have overlapped categories but also have unique categories. I believe it's more important to handle unique categories that are annotated in one dataset but not in the others. That's because unannotated objects in the other datasets will be regarded as background by mistake. It's better to have a solution to handle both overlapped and unique categories.

2. RoI head is able to refine various region proposals to the target-style boxes. Is it necessary to train a RPN as box generator that generates source-style boxes? It's better to compare the proposed solution with a only-RPN solution. That is, train a RPN (probably together with the two-stage architecture) with the target dataset and use it on different source datasets so that we don't need to train RPNs for different pairs of source-target datasets.

3. I'm not sure how to use the label translator to handle target datasets with more than one categories. Based on the experiments, it seems that the label translator only handles one category ('cyclist' in  Synscapes, Cityscapes, MVD, nuImages and Waymo). Probably, with the increase of categories, the label translator may introduce the noisy pseudo labels and negatively impact the performance.

4. Evaluations on widely used object detection datasets are missing. How about COCO, Objects365, and OpenImages? It seems that the experiments only include detection datasets for driving scenarios.

5. Based on Sect. 5.3, translation-mAP is not strongly correlated with downstream-mAP, and visually analyzing the translated labels is insufficient to judge label translation. Why is translation-mAP reported?

6. The proposed label translation seem to require training between every source and target datasets, which is costly to scale up.

**Questions:**

See Weakness.

---

> ### Author Response · Authors · 2023-11-15
> **Response to Reviewer MRBF**
>
> $\textbf{Dear Reviewer MRBF}$
>
> Thank you for your positive comments regarding the interesting research problem, the proposed usage of RoI head, and the good performance on various detectors and datasets.
>
> **Handling unique categories that are in one dataset but not in the others**
>
>
> This is a great point. Indeed, LGPL can handle the setting when two datasets have only partially overlapped categories, by translating the overlapped categories and leaving the others as is.
>
> To specifically translate unique classes, we note that LGPL is model/training-agnostic and can be applied with any other data augmentation or training algorithm. For example, if the target dataset has unique categories, we can use any off-the-shelf semi-supervised learning to learn the unique categories. If the source dataset has some unique categories, we can just drop those labels.
>
>
> **Necessity of training a RPN that generates source-style boxes**
>
> We interpret the reviewer's suggestion as: "Using the RPN trained on the target dataset so that the training/inference is not tied with any source dataset."
>
> We highlight that the pseudolabeling baseline is exactly training the RPN together with RoI on the target dataset. However, as shown in Table 1, pseudolabeling under-performs in the label translation problems, often with worse performance than "No translation".
>
>
> **Handle target datasets with more than one category**
>
>
> We apologize for the confusion. In fact, most of our experiments handled multiple categories, i.e., nuScenes $\rightarrow$ nuImages (10 classes), Synscapes $\rightarrow$ Cityscapes (7 classes), and Internal-Dataset $\rightarrow$ nuImages$^\dagger$ (3 classes). We provide the dataset statistics in Table 6 in the Appendix and the full class names in Section A.2. We have clarified in Section 5.1
>
>
> **Evaluation on other detection datasets**
>
> We expect our strategy to work on other common detection data sets like COCO and Object365 since our method and problem are not restricted to driving data. We focus on driving data sets because this is the real-world motivation of our work (e.g., see our Internal-Dataset) where the practice involves combining multiple data sets from different vendors [1] or mixing synthetic data [2]. We also emphasize that we considered four scenarios from seven different data sets and arrived at consistent results.
>
>
>
> **The role of translation-mAP**
>
>
> We absolutely agree with your assessment on the effectiveness of translation-mAP, which reflects the false intuition that visual inspection of translated labels might be sufficient to assess label quality. We believe it is important to disprove the false notion and emphasize the importance of downstream training when evaluating label quality.
>
>
> **Label translator is source-target specific**
>
>
>
> We agree that a universal label translator that would not require retraining on the source and the target datasets would be impactful and impressive. However, we emphasize that we are the first one that formally introduce the label translation problem. We feel it is necessary for this work to streamline the problem setup. Furthermore, implementing a universal label translator is non-trivial (e.g., off-the-shelf foundation models are not always effective as we show in Table 5).
> We hope to extend this framework to a universal translator in future work.
>
>
>
>
> [1] Yan Wang et al., Train in Germany, Test in The USA: Making 3D Object Detectors Generalize, Conference on Computer Vision and Pattern Recognition , 2020.
>
> [2] Viraj Uday Prabhu et al., Bridging the sim2real gap with CARE: Supervised detection adaptation with conditional alignment and reweighting. Transactions on Machine Learning Research,2023.

---

> > ### Author Response · Authors · 2023-11-21
> >
> > Dear Reviewer MRBF,
> >
> > Thank you for your insightful suggestions. We've addressed your concerns in our response and value your feedback on its effectiveness. Your contribution to refining our manuscript is greatly appreciated. We look forward to your reply before the rebuttal period ends. Thanks again for your time and consideration.
> >
> > Best regards,
> >
> > Authors

---

> > > ### Comment · Reviewer_MRBF · 2023-11-23
> > > **Thanks for your response**
> > >
> > > I thank the authors for their response. I feel most of my concerns have been addressed, but I still encourage the authors to add results on common object detection benchmarks (e.g., COCO), which might demonstrate the value of the proposed method in more general cases. Overall, I still remain positive about this paper, but given the weakness raised by the other reviewers (e.g., limited insights) and the missing results aforementioned, I'd like to keep my initial rating.

---

### Official Review · Reviewer_XHr3 · 2023-11-01

**Soundness:** 3 good
**Presentation:** 3 good
**Contribution:** 2 fair
**Rating:** 6
**Confidence:** 4

**Summary:**

This paper fuscous on object detection and tries to address the annotation mismatch issue among different datasets.
This paper first formally defines the label translation problem and proposes a taxonomy that characterizes the annotation mismatches across object detection datasets. In addition, this paper  introduces a simple yet effective label-guided pseudo-labeling (LGPL) approach. The proposed LGPL method extends the concept of pseudo-labeling by leveraging source dataset bounding boxes and class information for label translation. Comprehensive experiments and analysis on four translation scenarios across seven datasets validate the effectiveness of the proposed method.

**Strengths:**

+ This paper is well organized and written. The overall motivation is clear and convincing.
+ The research problem defined in this paper is interesting and practical.
+ Promising results are achieved compared to several baselines.

**Weaknesses:**

- Although the introduced research problem is interesting. The technical contributions of this paper is somewhat limited. It would be better to further highlight the novelty and technical contributions.
- In Figure 4, the proposed method is compared to an unsupervised domain adaptive object detection method and an unsupervised image domain adaptive method. In my understanding, the two domain adaptive methods don’t have any labels in the target domain. It is not strange the proposed method achieves superior performance, as it uses more annotations. I’m wondering why the proposed method is not compared to some weakly supervised cross-domain object detection methods, such as "H2FA R-CNN: Holistic and Hierarchical Feature Alignment for Cross-Domain Weakly Supervised Object Detection, CVPR’22", where all the image-level annotations are available.
- The class-wise threshold $\sigma_c$ is ad hoc. How to choose the value? Is it sensitive to the final results?

**Questions:**

Please refer the Weaknesses.

---

> ### Author Response · Authors · 2023-11-16
> **Response to Reviewer XHr3**
>
> $\textbf{Dear Reviewer XHr3}$
>
>
> Thank you for the positive comments regarding the practical importance of our problem, the presentation of our work, and the promising results achieved compared to the baselines.
>
>
> **Novelty and technical contributions**
>
> We revised the paper to clarify our novelty and the contributions in the Introduction and the Conclusion.
>
>
> To clarify our novelty and technical contributions:
>
> 1. **Novelty**: We are the first to introduce and formalize the label translation problem, to the best of our knowledge. We demonstrate that this problem is prevalent with an extensive study of several object detection datasets, by categorizing a set of four common types of annotation mismatches.
> 2. **Technical/Methodology**: We propose a simple but effective label translation algorithm, which, although motivated by pseudolabeling, expands upon this naive baseline. Unlike pseudolabeling, LGPL effectively leverages source dataset bounding boxes and class labels during label translation.
> 3. **Empirical**: We extensively validate our approach on seven datasets and four source-target combinations. Our approach *consistently* outperforms all the baselines in every setting. Importantly, we compare against strategies like supervised domain adaptation [1], demonstrating that our data-centric approach is a new alternative to the standard method for addressing image/label mismatch.
>
>
> **Comparison with supervised domain adaptive object detection method [1]**
>
> We apologize for the confusion on this experiment. In Figure 4, we indeed compare the proposed approach with supervised domain adaptation (SDA) [1]. Prior art [1] adopts popular unsupervised domain adaptation (UDA) approaches to the supervised settings, where both the source and the target labels are accessible. We adopt two variants S-DANN and S-CycConf from [1]. Note that we prefix the supervised version with "S".
>
> SDA approaches proposed in [1] directly align instance features given the source and the target labels, while cross-domain weakly supervised object detection (CD-WSOD) either aligns instance-level features via pseudolabels [2,3] or aligns image-level features [4]. We choose SDA as our baselines since SDA avoids noisy pseudolabels and is able to align instance-level features between the source and the target domain effectively.
>
> We appreciate the reviewer's careful review. We have fixed the submission to clarify the comparison in Figure 4 and added a brief background to the supervised domain adaptation in Section C.1.
>
>
>
> **Details of choosing class-conditional thresholds $\sigma_c$**
>
> Our revision clarifies this point in Appendix D.1.
>
> We choose $\sigma_c$ by quantizing the confidence score [5] for each class and applying it to all the label translators. Annotations falling into the last bin or with a confidence score lower than $0.001$ are dropped. In this way, the classes that are more challenging get the lower thresholds and vice versa. We leave further investigation of the class-conditional threshold to future research.
>
> Let $v_i$ be the value of the $i^{th}$ bin. We performs sensitivity test w.r.t. $\sigma_c$ on nuScenes $\rightarrow$ nuImages:
>
> | Label translator                  | Faster-RCNN Downstream-mAP |
> |-------------------|----------------|
> | LGPL with $\sigma_c = v_1$  | 42.6 $\pm$ 0.1               |
> | LGPL with $\sigma_c = v_2$  | 42.02               |
> | LGPL with $\sigma_c = v_5$  | 42.05               |
>
> From the experiments above, the choices of the class-conditional threshold do have impacts on the downstream-mAP but all of them still outperform the baselines.
>
>
> [1] Viraj Uday Prabhu et al., Bridging the sim2real gap with CARE: Supervised detection adaptation with conditional alignment and reweighting. Transactions on Machine Learning Research,2023.
>
> [2] Naoto Inoue et al., Cross-Domain Weakly-Supervised Object Detection through Progressive Domain Adaptation, CVPR2018
>
> [3] Shenxiong Ouyang et al., Pseudo-Label Generation-Evaluation Framework For Cross Domain Weakly Supervised Object Detection, ICIP 2021
>
> [4] Yunqiu Xu et al., H2FA R-CNN: Holistic and Hierarchical Feature Alignment for Cross-domain Weakly Supervised Object Detection, CVPR2022
>
>
> [5] Herbert A. Sturges, The choice of a class interval, Journal of the American Statistical Association 1926

---

> > ### Comment · Reviewer_XHr3 · 2023-11-20
> >
> > Thanks for the authors’ efforts in rebuttal.
> >
> > The authors clarify most of my concerns in the experiment setup and design choice (e.g., supervised domain adaptive object detection and $\sigma_c$).
> >
> > Despite the novelty regarding research problem formulation, I remain apprehensive about the overall technical contributions of this work. The proposed method is appreciated for its simplicity and effectiveness, but I feel the proposed method brings limited insights. Furthermore, as other reviewer pointed out, the research problem investigated is based on some strong hypotheses that limit its application.
> >
> > After carefully reading the response and other reviewers' comments, I would like to keep my rating.

---

> > > ### Author Response · Authors · 2023-11-21
> > >
> > > We sincerely appreciate your positive sentiment towards our work regarding its simplicity and effectiveness. We've revised the submission and conveyed our insights more clear. We would like to thank the reviewer again for the feedback as well as the active engagement in the rebuttal!
> > >
> > > Best regards,
> > >
> > > Author

---

### Author Response · Authors · 2023-11-16
**General response to all reviewers**

$\textbf{Dear all Reviewers}$

We thank the reviewers for their constructive suggestions! All reviewers remarked positively on (i) the significance of the problem we explore, and (ii) the strong empirical performance of our method, LGPL. We revised the paper according to the suggestions and highlighted the main differences.


We highlight the main points here:
1. **Handling partially overlapped class label set**: We apologize for any confusion. Our revision clarifies that LGPL can handle datasets with different class label sets by translating specifically the overlapping classes. Due to the data-centric property, LGPL easily combines with any post-processing or custom training algorithm for handling unique classes.
2. **Additional experimental results**: We provide the sensitivity test on the class-conditional threshold in response to Reviewer XHr3.
3. **Extension of our work**: We thank the reviewers for the suggestions to extend this work, e.g., resolving arbitrary translations or developing a universal label translator. We consider our streamlined version of addressing specifically bounding box mismatches as the necessary first step in this novel line of research.

For more details responses, please refer to the individual responses.

ICLR 2024 Conference Paper334 Authors

---

### Meta-Review · Area_Chair_SpSW · 2023-12-08

**Metareview:**

All the reviewers acknowledged the importance of the paper as it attempted to address (object detection) OD dataset mismatch issues, e.g. inconsistent class labels and bounding boxes. The paper formally defined the “label translation problem” and proposed a taxonomy that characterizes (four kinds of) annotation mismatches across OD datasets, and solved the mismatch problem by a proposed label-guided pseudo-labeling (LGPL) approach. Experimental results demonstrated on average 1.88 mAP and 2.65 AP (@75) improvements.

Most of the reviewers’ comments were addressed in the rebuttal quite smoothly: e.g. lack of technical contributions, why the proposed method is not compared to weakly supervised cross-domain OD methods, the necessity of a RPN head with a RoI head, handling unique categories that are in one but no in others, and potentially narrowing down the scope of the mismatch problem. The authors did a good job overall to address these and the reviewers were content with the rebuttal.

The AC recommends accept as all the reviewers unanimously voted for (weak) accept, and the paper is among the first to address an overlooked important problem in the field. Please incorporate the points clarified in the rebuttal process to the final version.

**Justification For Why Not Higher Score:**

The scope of the paper could have been bigger for the paper to have a higher score. Yes, the paper is among the first to address the annotation translation problem, but the it could be dealing with more complex situations (for now it's just word level overlap).

**Justification For Why Not Lower Score:**

The paper is among the first to propose the annotation translation problem, which is an important data preparation/cleaning pipeline for large-scale training. It was probably done in an ad-hoc way before, and it's nice to formulate and solve it in a principal way.

---

### Decision · Program_Chairs · 2024-01-16

Accept (poster)